# Comparison of the dynamics of exoskeletal-assisted and unassisted locomotion in an FDA-approved lower extremity device: Controlled experiments and development of a subject-specific virtual simulator

**Vishnu D. Chandran[1], Sanghyun Nam[1], David Hexner[2], William A. Bauman[3,4], Saikat Pal [1,5] ***

**1** Department of Biomedical Engineering, New Jersey Institute of Technology, Newark, New Jersey, United States of America, **2** ReWalk Robotics, Yokneam, Israel, **3** James J. Peters Veterans Affairs Medical Center, Bronx, New York, United States of America, **4** Department of Medicine and Rehabilitation & Human Performance, Icahn School of Medicine at Mount Sinai, New York, New York, United States of America, **5** Department of Electrical and Computer Engineering, New Jersey Institute of Technology, Newark, New Jersey, United States of America

* pal@njit.edu

## Abstract

Robotic exoskeletons have considerable, but largely untapped, potential to restore mobility in individuals with neurological disorders, and other conditions that result in partial or complete immobilization. The growing demand for these devices necessitates the development of technology to characterize the human-robot system during exoskeletal-assisted locomotion (EAL) and accelerate robot design refinements. The goal of this study was to combine controlled experiments with computational modeling to build a virtual simulator of EAL. The first objective was to acquire a minimum empirical dataset comprising human-robot kinematics, ground reaction forces, and electromyography during exoskeletal-assisted and unassisted locomotion from an able-bodied participant. The second objective was to quantify the dynamics of the human-robot system using a subject-specific virtual simulator reproducing EAL compared to the dynamics of normal gait. We trained an able-bodied participant to ambulate independently in a Food and Drug Administration-approved exoskeleton, the ReWalk P6.0 (ReWalk Robotics, Yoknaem, Israel). We analyzed the motion of the participant during exoskeletal-assisted and unassisted walking, sit-to-stand, and stand-to-sit maneuvers, with simultaneous measurements of (i) three-dimensional marker trajectories, (ii) ground reaction forces, (iii) electromyography, and (iv) exoskeleton encoder data. We created a virtual simulator in OpenSim, comprising a whole-body musculoskeletal model and a full-scale exoskeleton model, to determine the joint kinematics and moments during exoskeletal-assisted and unassisted maneuvers. Mean peak knee flexion angles of the human subject during exoskeletal-assisted walking were 50.1˚ ± 0.6˚ (left) and 52.6˚ ± 0.7˚ (right), compared to 68.6˚ ± 0.3˚ (left) and 70.7˚ ± 1.1˚ (right) during unassisted walking. Mean peak knee extension moments during exoskeletal-assisted walking were 0.10 ± 0.10 Nm/kg (left) and 0.22 ± 0.11 Nm/kg (right), compared to 0.64 ± 0.07 Nm/kg (left) and 0.73 ±

**Data Availability Statement:** All relevant data are available on Figshare: https://doi.org/10.6084/m9.figshare.21967874.v1.

**Funding:** WAB and SP received support from the Department of Veterans Affairs (https://www.va.gov/) grant VA RR&D # 1 I01 RX003561-01A2. The funders had no role in study design, data collection and analysis, decision to publish, or preparation of the manuscript.

**Competing interests:** The authors have declared that no competing interests exist.

0.10 Nm/kg (right) during unassisted walking. This work provides a foundation for parametric studies to characterize the effects of human and robot design variables, and predictive modeling to optimize human-robot interaction during EAL.

## Introduction

Wearable robotic exoskeletons have considerable potential to transform the people's lives by reducing the metabolic cost of walking [1–3], carrying heavy loads [4, 5], or augmenting human performance to conduct strenuous tasks over extended periods of time [6, 7]. The need for such assistive devices is particularly profound in restoring mobility in individuals with neurological disorders, including those of spinal cord injury, stroke, traumatic brain injury, and multiple sclerosis. Wearable robotic exoskeletons for rehabilitation of individuals with neurological disorders are relatively new. The ReWalk (ReWalk Robotics, Yoknaem, Israel) was the first lower extremity device to receive Food and Drug Administration (FDA) approval in 2014, followed by the Ekso (Ekso Bionics, Richmond, CA) and Indego (Parker Hannifin, Cleveland, OH) in 2016, and the Keeogo (B-temia Inc., Quebec City, Canada) in 2020. Prior studies have demonstrated the ability of individuals with neurological disorders who have partial impairments to those who are completely non-ambulatory to walk independently in robotic exoskeletons [8–19]. Exoskeletal-assisted locomotion (EAL) has been shown to improve functional and motor recovery [8], mobility [9, 10, 15, 16], chronic pain [20], muscle spasticity [20–22], cardiovascular health [13, 23], bowel function [24], bladder function [21, 22], and quality of life [21]. These studies highlight the growing demand for wearable robotic exoskeletons to improve physical and psychological health, employment opportunities, and community integration in persons with neurological disorders.

The growing demand for wearable robotic exoskeletons for rehabilitation of individuals with neurological disorders has exposed two gaps in knowledge. First, the dynamics (comprising joint kinematics and joint moments) of the human-robot system during EAL in FDA-approved devices are not well-understood. The human-robot entity during EAL represents a complex dynamic system, and parsing out the contributions of the human from the robot, together with their interaction effects, requires a minimum empirical dataset that includes human-robot kinematics, ground reaction forces, and electromyography (EMG). Such a dataset from EAL in an FDA-approved device does not exist. Kim and colleagues quantified the dynamics of the human-robot system during EAL in a device designed to augment human performance in industrial settings [25]; however, this exoskeleton is not FDA-approved for rehabilitation of individuals with neurological disorders.

It is important to study the dynamics of the human-robot system during EAL in FDA-approved devices because individuals with neurological disorders only have access to FDA-approved devices. Clinics and medical centers only use FDA-approved exoskeletons for rehabilitation of patients with neurological disorders. FDA approval is required to prescribe an exoskeleton for in home use by individuals with neurological disorders. Even in research settings, many institutional review boards require FDA approval for an exoskeleton to be used by participants with neurological disorders. These restrictions are due to the safety and regulatory concerns associated with the use of non-FDA approved devices, and highlight the importance of studying the dynamics of the human-robot system during EAL in FDA-approved devices. Previous studies with FDA-approved exoskeletons for neurological disorders have reported joint kinematics [10, 26–30], foot and ground reaction forces [14, 29, 30], and EMG [10, 17,

31, 32]. To the best of our knowledge, no prior study has reported joint moments of the human-robot system during EAL in an FDA-approved device. Joint moments are important because our understanding of the dynamics of a system is incomplete without joint moments. In dynamics, a moment of a force (for example, from a muscle) is a measure of its tendency to rotate a limb about its joint axis. Quantifying joint moments is mathematically complex since it requires solving equations of motions with subject-specific anthropometry, joint kinematics, and external forces as inputs. Furthermore, a minimum empirical dataset comprising joint kinematics, ground reaction forces, and EMG from a single session of controlled experiments with and without an FDA-approved exoskeleton is not available in the literature.

The second gap in knowledge exposed by the growing demand for wearable robotic exoskeletons is that existing technology to accelerate design refinements and improve human experience in these devices is extremely limited. It is not ethically feasible, and would undoubtedly be prohibitively expensive, to conduct a large number of human experiments to characterize the effects of an endless number of robot design variables, control strategies, and human parameters on the human-robot system during EAL. Computational simulation is a viable alternative to perform such parametric studies. Prior studies have used computational simulations of the human-robot system to optimize exoskeleton design [33–37], test control strategies [38–41], minimize the metabolic cost of locomotion [42–44], optimize assistance for pathological gait [45, 46], and study human-robot interaction [47–52]. However, no prior study has simulated EAL in an FDA-approved exoskeleton.

The goal of this study was to combine controlled experiments with computational modeling to build a virtual simulator of EAL in an FDA-approved lower extremity exoskeleton, the ReWalk P6.0. The first objective of this study was to acquire a minimum empirical dataset comprising human-robot kinematics, ground reaction forces, and EMG during exoskeletal-assisted and unassisted locomotion. The second objective of this study was to quantify the dynamics of the human-robot system using a subject-specific virtual simulator reproducing EAL compared to the dynamics of normal gait.

## Methods

### Participant recruitment

An able-bodied male (age 41 years, male, height 1.76 m, mass 89.4kg) was recruited for study participation. The participant was informed on all aspects of the study and signed informed consent in accordance with the policies of our Institutional Review Board. The Institutional Review Board at New Jersey Institute of Technology approved this study. The approval number is 2008001531R001.

### Exoskeletal-assisted locomotion training

The participant was trained to perform sit-to-stand, stand-to-sit, and walking maneuvers in the ReWalk P6.0. Prior to initiating training, the exoskeleton was adjusted to fit the anthropometry of the participant, including pelvic band size, thigh leg length, shank leg length, knee bracket position, foot plate size, and ankle dorsiflexion setting [53]. Walking in the ReWalk P6.0 follows a standard procedure. To initiate walking in the ReWalk P6.0, the user starts in the standing position with the hand crutches on each side. The user activates the "Walk" mode through a controller watch, which is confirmed by a single beep. To perform the device-assisted gait maneuver, the user unweights each side in an alternating manner—that is, the right leg is unloaded by leaning the torso to the left, which is followed by leaning the torso to the right, which unloads the left leg, and so forth. The action of the user swaying from side-to-side and, thereby, unloading the contralateral leg, activates the hip and knee motors to initiate the

stepping motion. The next several steps are a repetition of the back-and-forth swaying of the torso to unload the trailing leg to permit the stepping action to continue.

The sit-to-stand and stand-to-sit maneuvers in the ReWalk P6.0 follow standard procedures. To initiate the sit-to-stand maneuver, a user places the hand crutches posterior to the hip while seated. Then, the trainer activates the "Stand" mode through a controller, which is confirmed by a single beep. The hip motors flex the torso (~9°) to position the user's upper body forward while they load the crutches. The user has to hold this position for 3 seconds, after which time the exoskeleton beeps thrice and activates the hip and knee motors to complete the sit-to-stand maneuver. Next, the stand-to-sit maneuver in the ReWalk P6.0 follows a similar procedure. A user places the hand crutches posterior to the hip while standing. Then, the trainer activates the "Sit" mode, which is confirmed by a single beep. The user gets into position by leaning backward (~6°) and loading the crutches. The user has to hold this position for 3 seconds, after which time the exoskeleton beeps thrice and activates the hip and knee motors to complete the stand-to-sit maneuver. By the end of the fourth 1-hour session, the participant was able to perform the sit-to-stand, stand-to-sit, and walking maneuvers in the exoskeleton independent of requiring any assistance.

## Motion capture experiments during exoskeletal-assisted and unassisted locomotion

We analyzed the participant during exoskeletal-assisted and unassisted locomotion from a single motion capture session, including simultaneous measurements of three-dimensional (3-D) marker trajectories, ground reaction forces, EMG, and exoskeleton encoder data. A 16-camera motion capture system (Vicon V8 and Nexus, Vicon Motion Systems, Oxford, UK) was used to track retro-reflective markers at 100 Hz. The markers were placed on the participant and the exoskeleton to capture the position and orientation of all the human body and exoskeleton segments (Fig 1A and 1B). The markers were placed on the participant according to the Conventional Gait Model 2.5 template in Vicon Nexus [54]. Markers were placed on the exoskeleton based on a custom template (Fig 1A and 1B). Next, we recorded ground reaction forces from each foot using four overground force plates (Bertec Corp., Columbus, OH) sampled at 2000 Hz. The force data were filtered using a low-pass, fourth-order Butterworth filter with a cutoff frequency of 15 Hz.

We recorded muscle EMG measurements from both legs using a 16-channel surface system (Trigno$^{TM}$, Delsys Inc., Natick, MA) sampled at 2000 Hz. We measured EMG from rectus femoris, vastus lateralis, vastus medialis, semitendinosus, biceps femoris, gastrocnemius medialis, soleus, and tibialis anterior muscles using established protocols [56, 57]. The participant's mean resting EMG was determined from unassisted standing trials with the participant instructed to remain stationary in a neutral pose with their muscles relaxed. The mean resting EMG value was subtracted from their raw EMG values from the locomotion trials to offset the data to zero. We filtered the EMG data using a fourth-order high-pass filter (30 Hz) to remove motion artifact and then full-wave rectified and filtered with a fourth-order Butterworth low-pass filter (6 Hz) to obtain the linear envelope of muscle activation [58]. The filtered EMG data were normalized to muscle-specific activations obtained from maximum voluntary contraction trials. To quantify muscle-specific activations from maximum voluntary contraction trials, a participant performed 5 trials of isometric muscle contractions for each muscle group to elicit maximum activation of the specific muscles [58]. For example, to elicit maximum activation of the quadriceps muscles, a participant sat on a chair with the knee at approximately 80° of flexion and was instructed to extend their knee against the resistance of the tester. Verbal encouragement was given to the participant to try to improve the effort with each trial. The

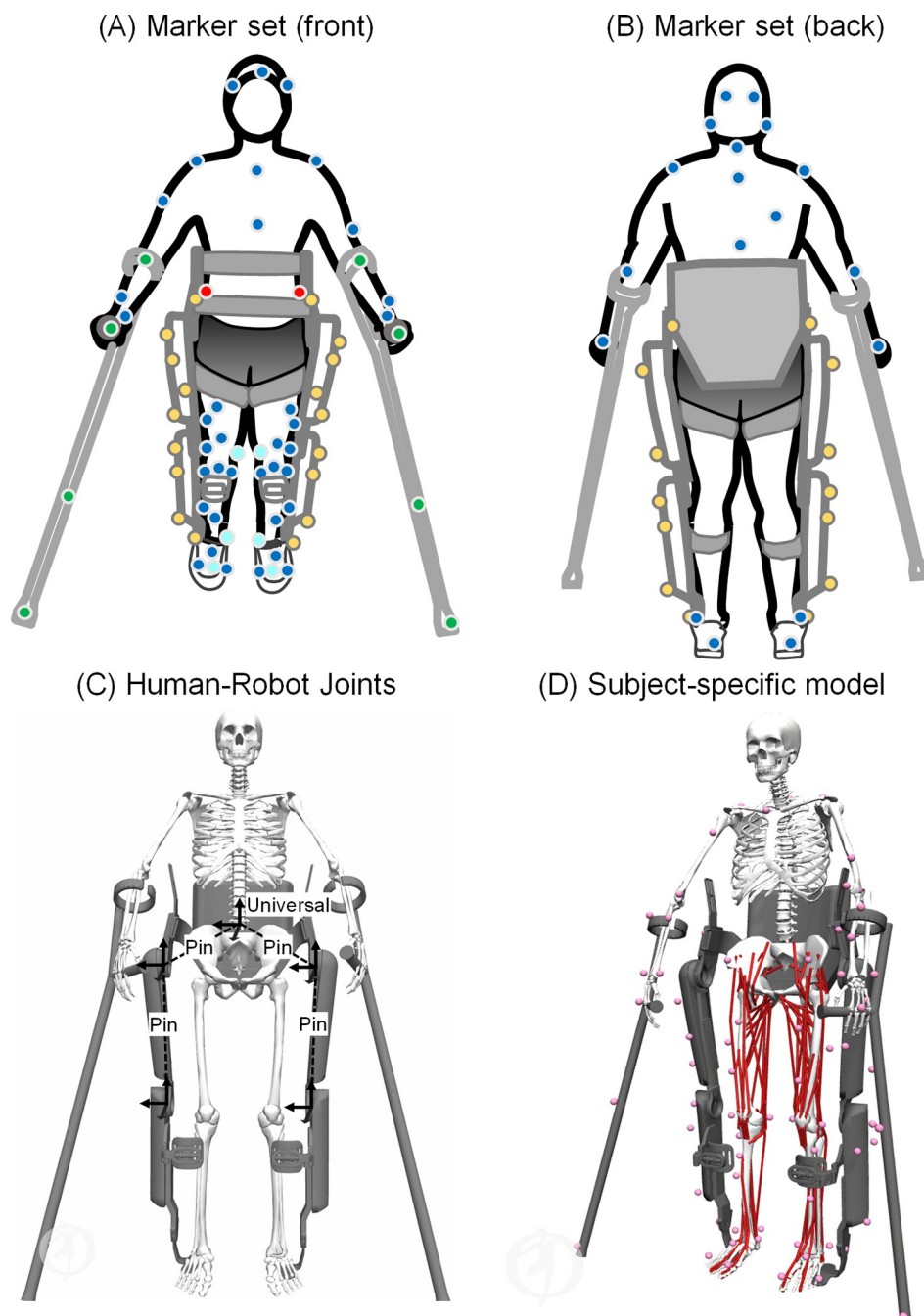

**Fig 1.** (A, B) Placement of retro-reflective markers on the human (blue, cyan, and red), exoskeleton (yellow), and crutches (green) to track the position and orientation of all the segments of the human-robot system. The blue and cyan markers represent the Conventional Gait Model (CGM) 2.5 marker set, with the cyan markers removed after initial calibration. The red markers are offset markers used to locate occluded anatomical landmarks when the participant is in the exoskeleton. During post-processing, the occluded anatomical landmarks are reconstructed using these offset markers to complete the CGM 2.5 marker set. Next, we created a custom template to track the different segments of the exoskeleton and crutches. (C) The human-robot model in the virtual simulator comprises a universal joint (six degrees of freedom) at the pelvic band to anchor the exoskeleton to the participant, and a pin joint (one degree of freedom) each at the hips and knees of the exoskeleton. The location of the different joint axes are shown. (D) A generic musculoskeletal model [55] was scaled to the participant's anthropometry.

peak EMG value from all 5 trials was assigned as a muscle's maximum activation. Similar isometric muscle contraction trials were performed for the hamstrings, ankle plantarflexion, and ankle dorsiflexion muscle groups. In addition, we recorded exoskeleton encoder data from all motion capture trials, including timestamps corresponding to the activation of the motors and motor encoder angles. These data were synchronized with the 3-D marker trajectory, ground reaction force, and EMG data.

## Motion capture of exoskeletal-assisted and unassisted walking

We acquired 3-D motion data while the participant walked back and forth over a 5.0 x 2.5 m even instrumented walkway with and without the exoskeleton. The participant walked 10 times back and forth with the exoskeleton at the participant's preferred speed (0.47 ± 0.03 m/s), and 10 times unassisted at self-selected speed (1.11 ± 0.07 m/s). From this dataset, only successful trials were included for further analysis. A walking trial was determined as successful if the foot placements were entirely on single force plates, and there were no missing marker, ground reaction force, and EMG data. Additional criteria for a successful trial during exoskeletal-assisting walking was that the hand crutches were placed away from the force plates so as not to interfere with the ground reaction forces from the feet, and there were no missing exoskeleton encoder data. Based on these criteria, we obtained six successful trials each of exoskeletal-assisted and unassisted walking.

## Motion capture of exoskeletal-assisted and unassisted sit-to-stand and stand-to-sit maneuvers

We acquired 3-D motion data while the participant performed sit-to-stand and stand-to-sit maneuvers with and without the exoskeleton. The participant used an armless piano bench with seat height of 43 cm positioned on the overground walkway that enabled each foot to be placed on a force plate during the entire duration of a maneuver. The participant performed the sit-to-stand and stand-to-sit maneuvers 10 times each with and without the exoskeleton. From this dataset, only successful trials were included for further analysis. A trial was determined as successful if the foot placements were entirely on single force plates and there were no missing marker, ground reaction force, and EMG data. Additional criteria for a successful trial during exoskeletal-assisting sit-to-stand/stand-to-sit were that the hand crutches were placed away from the force plates so as not to interfere with the ground reaction forces from the feet and there were no missing exoskeleton encoder data. Based on these criteria, we obtained four and five successful trials of exoskeletal-assisted and unassisted sit-to-stand maneuvers, respectively, and five successful trials each of exoskeletal-assisted and unassisted stand-to-sit maneuvers.

## Virtual simulator reproducing exoskeletal-assisted and unassisted locomotion

We developed a virtual simulator in OpenSim [59] to reproduce exoskeletal-assisted and unassisted walking, sit-to-stand, and stand-to-sit maneuvers (Fig 1C and 1D). We integrated a previously published human musculoskeletal model [55] with a full-scale model of the ReWalk P6.0 exoskeleton. The musculoskeletal model comprised 24 segments and 37 degrees of freedom (DoFs): seven in each leg, six at the pelvis, three at the torso, and seven in each arm [55]. The seven DoFs in each leg included three DoFs at the ball-and-socket hip joint, a one DoF coupled knee mechanism with translations of the tibia and patella prescribed by the knee flexion angle, and one DoF revolute joints at the ankle, subtalar, and metatarsal joints. A six DoF universal joint at the pelvis was used to describe the pose of the musculoskeletal model with respect to the global origin. The three DoFs in the torso and upper body included a spherical

joint connecting the torso to the pelvis. The seven DoFs in each arm included three DoFs at the ball-and-socket shoulder joint, one DoF revolute joint at the elbow, one DoF revolute joint between the radius and the ulna, and two DoFs universal joint between the radius and the hand for wrist flexion-extension and radial-ulnar deviation. The full-scale model of the exoskeleton comprised seven segments and four DoFs (Fig 1C). The four DoFs included a single DoF pin joint each at the hips and knees of the exoskeleton. The ankle joints in the exoskeleton were welded to the lower leg segments. Next, human-robot interactions were modeled using an additional six DoF universal joint at the pelvic band to anchor the exoskeleton to the participant (Fig 1C). The musculoskeletal model included 80 Hill-type muscle-tendon units [60]. The Hill-type muscle-tendon units captured the force-length-velocity properties of the lower extremity muscles, with muscle geometry and architecture based on adult cadaver data [61].

We adapted a previously published computational framework in OpenSim [55, 59] to determine the dynamics of exoskeletal-assisted and unassisted walking, sit-to-stand, and stand-to-sit maneuvers. We scaled the generic musculoskeletal model to match the mass and segment lengths of the participant. We determined joint kinematics by performing Inverse Kinematics (IK) analyses in OpenSim. IK solves for kinematics by minimizing error between the experimentally measured marker positions and the corresponding markers on the human-robot model. We performed Inverse Dynamics (ID) analyses to compute net human-robot joint torques. For ID analyses, we simplified the human-robot model by incorporating the mass and inertial properties of the exoskeleton with the musculoskeletal segments, similar to previous studies [62, 63]. Exoskeleton masses were added to the respective segments in the musculoskeletal model. The centers of mass of the segments in the musculoskeletal model were moved to the locations corresponding to the combined human-robot segments.

## Data analysis and statistical methods

We compared hip, knee, and ankle flexion-extension angles from each leg during exoskeletal-assisted and unassisted walking, sit-to-stand, and stand-to-sit maneuvers. For exoskeletal-assisted maneuvers, we determined the hip and knee flexion-extension angles from the robot using IK and exoskeleton encoder data; the RMS errors between these two methods were minimal (the worst case was 3.3˚), as such only joint angles from IK are presented in this study. The joint angles from multiple trials were averaged for each leg. We compared the vertical, anterior-posterior (AP), and medial-lateral (ML) ground reaction forces from each leg during exoskeletal-assisted and unassisted walking, sit-to-stand, and stand-to-sit maneuvers. The ground reaction forces from EAL were normalized to the combined weight of the participant and the exoskeleton [30]; forces from unassisted locomotion were normalized to the participant's body weight. Ground reaction forces from multiple trials were averaged for each leg. Next, we compared normalized EMG from the eight lower extremity muscles from each leg during exoskeletal-assisted and unassisted walking, sit-to-stand, and stand-to-sit maneuvers. Normalized EMG from multiple trials were averaged for each leg. Finally, we compared hip, knee, and ankle moments from each leg during exoskeletal-assisted and unassisted walking, sit-to-stand, and stand-to-sit maneuvers. The joint moments from EAL were normalized to the combined mass of the participant and the exoskeleton; joint moments from unassisted locomotion were normalized to the participant's mass. The joint moments from multiple trials were averaged for each leg.

## Results

### Exoskeletal-assisted and unassisted walking

The virtual simulator reproduced exoskeletal-assisted and unassisted walking within acceptable tolerances, with average (±1 SD) RMS errors between experiment and simulator markers

being 1.30 (± 0.05) cm and 1.20 (± 0.02) cm for exoskeletal-assisted and unassisted trials, respectively. Acceptable tolerance is less than 2 cm average RMS error between experiment and simulator markers, per OpenSim's best practices [64, 65]. Our segment coordinate frames were obtained with reference to a single global coordinate frame. The initial values of the joint kinematics from the different segment coordinate frames were based on this single global coordinate frame. This provided a direct comparison of joint kinematics with and without the exoskeleton. We observed differences in joint kinematics between exoskeletal-assisted and unassisted walking (Fig 2). The participant's hips were more flexed during exoskeletal-assisted walking compared to unassisted walking (Fig 2A and 2B). Mean peak hip extension angles of the human during exoskeletal-assisted walking were 3.1˚ ± 1.9˚ (left) and 0.2˚ ± 0.9˚ (right), compared to 27.2˚ ± 1.2˚ (left) and 25.1˚ ± 1.0˚ (right) during unassisted walking (Fig 2A and 2B). After subtracting the offset angles between the segment coordinate axes, average absolute difference between human and robot hip flexion-extension angles were 1.5˚ ± 1.4˚ (left) and 1.9˚ ± 1.0˚ (right). Next, mean peak knee flexion angles of the participant during exoskeletal-assisted walking were 50.1˚ ± 0.6˚ (left) and 52.6˚ ± 0.7˚ (right), compared to 68.6˚ ± 0.3˚ (left) and 70.7˚ ± 1.1˚ (right) during unassisted walking (Fig 2C and 2D). After subtracting the offset angles between the segment coordinate axes, average absolute difference between participant and robot mean knee flexion-extension angles were 1.7˚ ± 1.1˚ (left) and 0.8˚ ± 0.5˚ (right). Mean peak ankle plantarflexion angles of the participant during exoskeletal-assisted walking were 6.9˚ ± 2.8˚ (left) and 6.7˚ ± 3.9˚ (right), compared to 13.9˚ ± 2.5˚ (left) and 19.5˚ ± 1.9˚ (right) during unassisted walking (Fig 2E and 2F). Peak plantarflexion during exoskeletal-assisted walking occurred in early stance during braking, in contrast to peak plantarflexion at the end of propulsion (toe-off) during unassisted walking. We observed a minimal range of plantarflexion (5.3˚-10.5˚) in the propulsion phase during exoskeletal-assisted walking, compared to a larger range (27.1˚- 30.3˚) during unassisted walking.

We observed differences in normalized ground reaction forces between exoskeletal-assisted and unassisted walking, especially during the braking and propulsion phases (Fig 3). Mean peak vertical ground reaction forces during braking were 0.91 ± 0.08 BW (left) and 0.89 ± 0.06 BW (right) during exoskeletal-assisted walking, compared to 1.09± 0.02 BW (left) and 1.05 ± 0.05 BW (right) during unassisted walking (Fig 3A and 3B). Mean peak vertical ground reaction forces during propulsion were 0.98 ± 0.02 BW (left) and 1.02 ± 0.02 BW (right) during exoskeletal-assisted walking, compared to 1.04 ± 0.04 BW (left) and 1.02 ± 0.02 BW (right) during unassisted walking (Fig 3A and 3B). Mean peak posterior ground reaction forces during braking were 0.08 ± 0.04 BW (left) and 0.08 ± 0.02 BW (right) during exoskeletal-assisted walking, compared to 0.17 ± 0.01 BW (left) and 0.17 ± 0.04 BW (right) during unassisted walking (Fig 3C and 3D). Mean peak anterior ground reaction forces during propulsion were 0.06 ± 0.02 BW (left) and 0.08 ± 0.06 BW (right) during exoskeletal-assisted walking, compared to 0.18 ± 0.01 BW (left) and 0.19 ± 0.01 BW (right) during unassisted walking (Fig 3C and 3D). Mean peak lateral ground reaction forces during braking were 0.07 ± 0.02 BW (left) and 0.06 ± 0.01 BW (right) during exoskeletal-assisted walking, compared to 0.09 ± 0.01 BW (left) and 0.09 ± 0.01 BW (right) during unassisted walking (Fig 3E and 3F). Mean peak lateral ground reaction forces during propulsion 0.06 ± 0.01 BW (left) and 0.07 ± 0.01 BW (right) during exoskeletal-assisted walking, compared to 0.08 ± 0.01 BW (left) and 0.10 ± 0.01 BW (right) during unassisted walking (Fig 3E and 3F). The decreased ground reaction forces during the braking and propulsion phases of exoskeletal-assisted walking were supported by EMG data (Fig 4). The tibialis anterior muscle, used for braking, was less active during the braking and terminal swing (to prepare for braking) phases during exoskeletal-assisted compared to unassisted walking (Fig 4O and 4P). The gastrocnemius medialis and soleus muscles, used for

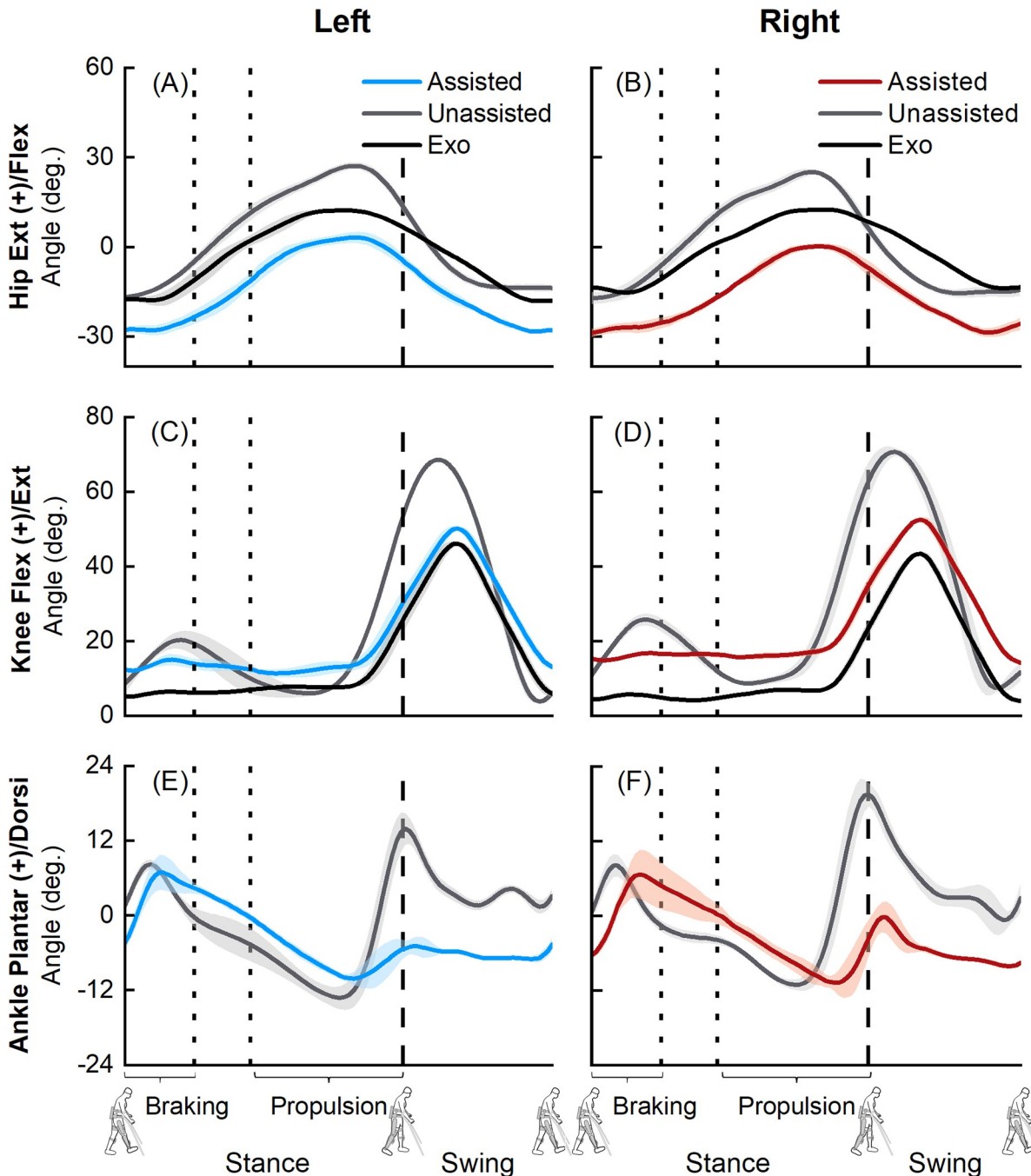

**Fig 2.** Average (±1 SD) hip (A, B), knee (C, D), and ankle (E, F) joint angles from six exoskeletal-assisted (human: blue for left leg, red for right leg; robot: black) and six unassisted (human: grey) walking trials. The offset in angles between the participant (blue or red) and the robot (black) was due to different definitions of coordinate axes for each rigid body. The dashed vertical lines represent toe-off.

propulsion, were less active during the propulsion phase during exoskeletal-assisted compared to unassisted walking (Fig 4K–4N).

We observed differences in joint moments between exoskeletal-assisted and unassisted walking (Fig 5). Mean peak hip flexion moments of the participant during exoskeletal-assisted walking 0.23 ± 0.05 Nm/kg (left) and 0.35 ± 0.06 Nm/kg (right), compared to 0.65 ± 0.07 Nm/

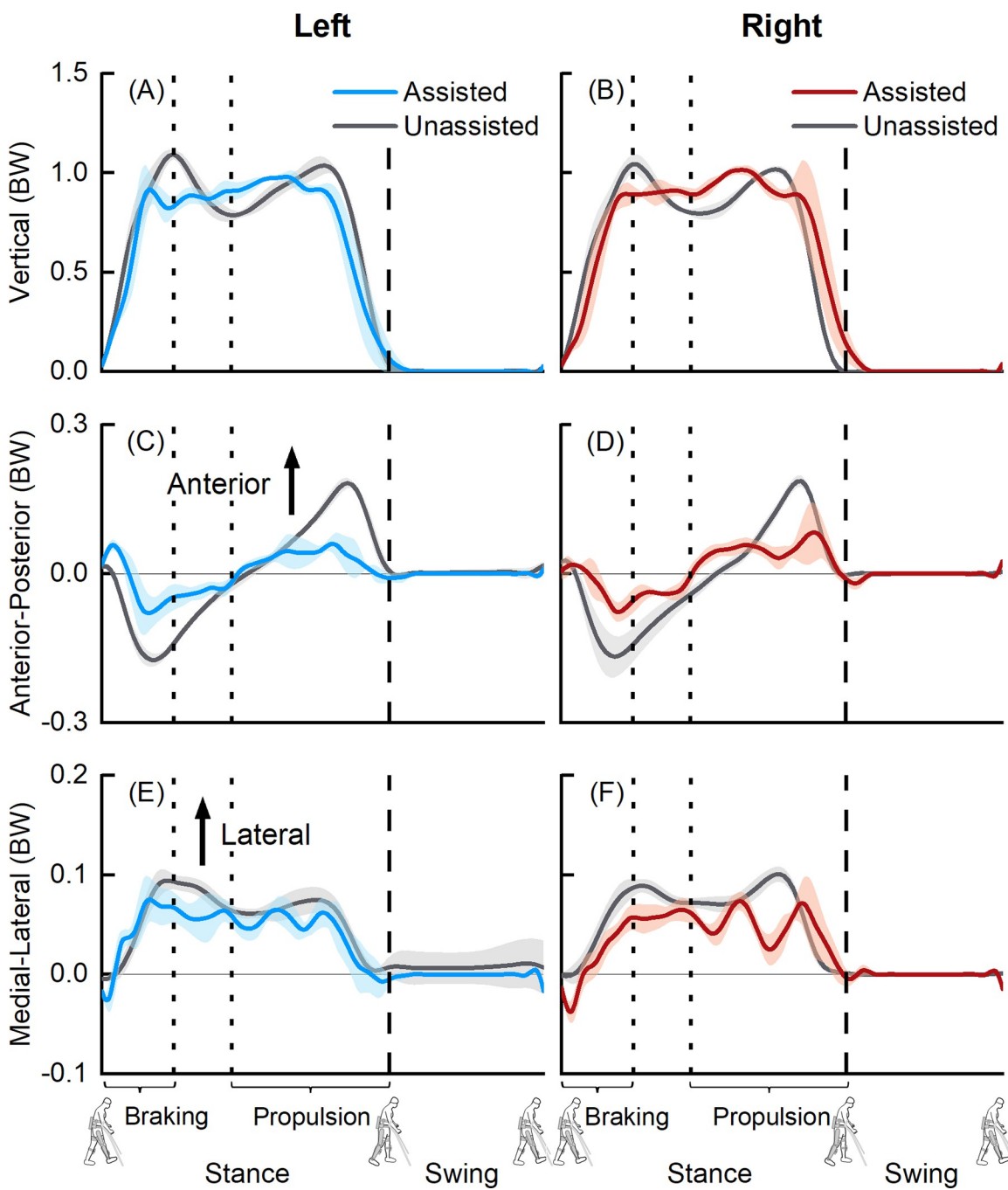

**Fig 3.** Average (±1 SD) vertical (A, B), anterior-posterior (C, D) and medial-lateral (E, F) ground reaction forces from six exoskeletal-assisted and six unassisted walking trials. The ground reaction forces from EAL were normalized to the combined weight of the participant and the exoskeleton; forces from unassisted locomotion were normalized to the participant's body weight (BW). The dashed vertical lines represent toe-off.

kg (left) and 0.64 ± 0.09 Nm/kg (right) during unassisted walking (Fig 5A and 5B). Mean peak knee extension moments of the participant during exoskeletal-assisted walking were 0.10 ± 0.10 Nm/kg (left) and 0.22 ± 0.11 Nm/kg (right), compared to 0.64 ± 0.07 Nm/kg (left) and 0.73 ± 0.10 Nm/kg (right) during unassisted walking (Fig 5C and 5D). Mean peak ankle

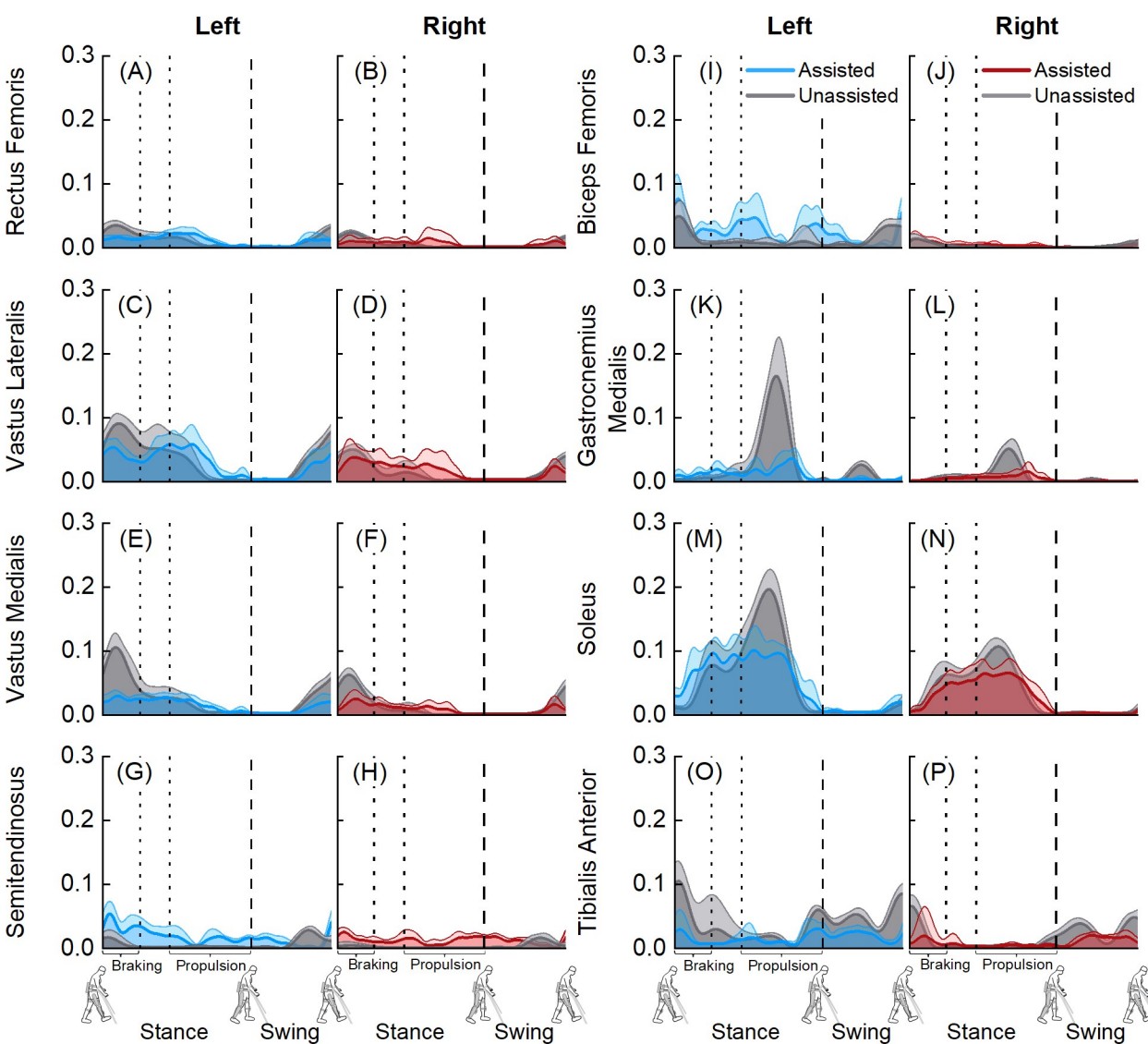

**Fig 4. Average (+1 SD) normalized electromyography (EMG) data from six exoskeletal-assisted and six unassisted walking trials.** The EMG data were normalized using muscle-specific maximum voluntary contraction values. The dashed vertical lines represent toe-off.

plantarflexion moments during exoskeletal-assisted walking were 0.80 ± 0.12 Nm/kg (left) and 0.78 ± 0.19 Nm/kg (right), compared to 1.29 ± 0.04 Nm/kg (left) and 1.12 ± 0.03 Nm/kg (right) during unassisted walking (Fig 5E and 5F).

## Exoskeletal-assisted and unassisted sit-to-stand

The virtual simulator reproduced exoskeletal-assisted and unassisted sit-to-stand within acceptable tolerances, with average (±1 SD) RMS errors between experiment and simulator markers being 1.50 (± 0.04) cm and 1.90 (± 0.05) cm for exoskeletal-assisted and unassisted trials, respectively. Exoskeletal-assisted sit-to-stand in the ReWalk P6.0 follows a standard procedure that is different from unassisted sit-to-stand, including different temporal events and phases. Comparison of exoskeletal-assisted and unassisted sit-to-stand is included as supplemental information, including joint kinematics (S1 Fig), ground reaction forces (S2 Fig), EMG (S3 Fig), and joint moments (S4 Fig).

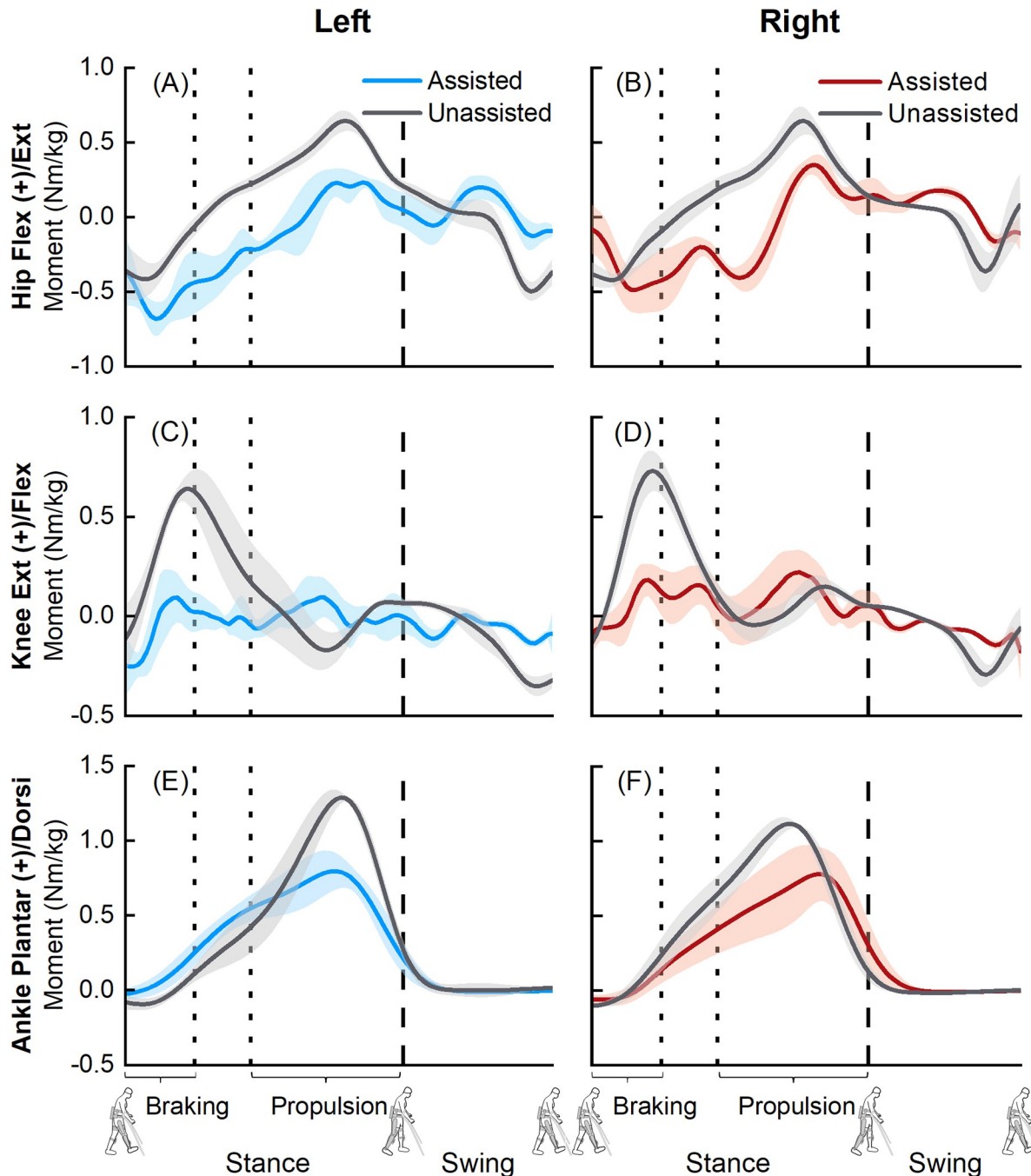

**Fig 5.** Average (±1 SD) hip (A, B), knee (C, D), and ankle (E, F) joint moments from six exoskeletal-assisted (human: blue for left leg, red for right leg) and six unassisted (human: grey) walking trials. The joint moments from EAL were normalized to the combined mass of the participant and the exoskeleton; joint moments from unassisted locomotion were normalized to the participant's mass. The dashed vertical lines represent toe-off.

## Exoskeletal-assisted and unassisted stand-to-sit

The virtual simulator reproduced exoskeletal-assisted and unassisted stand-to-sit within acceptable tolerances, with average (±1 SD) RMS errors between experiment and simulator markers being 1.40 (± 0.06) cm and 1.90 (± 0.12) cm for exoskeletal-assisted and unassisted

trials, respectively. Exoskeletal-assisted stand-to-sit in the ReWalk P6.0 follows a standard procedure that is different from unassisted stand-to-sit, including different temporal events and phases. Comparison of exoskeletal-assisted and unassisted stand-to-sit is included as supplemental information, including joint kinematics (S5 Fig), ground reaction forces (S6 Fig), EMG (S7 Fig), and joint moments (S8 Fig).

## Discussion

The goal of this study was to combine controlled experiments with computational modeling to build a virtual simulator of EAL in the ReWalk P6.0, an FDA-approved lower extremity exoskeleton for rehabilitation of patients with neurological disorders. The first objective of this study was to acquire a minimum empirical dataset comprising human-robot kinematics, ground reaction forces, and EMG during exoskeletal-assisted and unassisted locomotion. Here, we present a minimum empirical dataset from a single session of controlled experiments with an able-bodied participant performing exoskeletal-assisted and unassisted walking, sit-to-stand, and stand-to-sit maneuvers. Our results provide a direct comparison of joint kinematics (Fig 2, S1 and S5 Figs), ground reaction forces (Fig 3, S2 and S6 Figs), and EMG (Fig 4, S3 and S7 Figs) during exoskeletal-assisted and unassisted locomotion in an FDA-approved exoskeleton. The second objective of this study was to quantify the dynamics of the human-robot system using a subject-specific virtual simulator reproducing EAL, and to compare with the dynamics of unassisted locomotion. The subject-specific virtual simulator reproduced EAL within acceptable tolerances (average±SD RMS errors of 1.20±0.02 to 1.30±0.05 cm for walking, 1.50±0.04 to 1.90±0.05 cm for sit-to-stand, and 1.40±0.06 to 1.90±0.12 cm for stand-to-sit maneuvers). Our results provide a direct comparison of joint moments during exoskeletal-assisted and unassisted locomotion in an FDA-approved exoskeleton (Fig 5, S4 and S8 Figs).

We are not aware of any prior literature that has provided a minimum empirical dataset to quantify the dynamics of the human-robot system during EAL in any FDA-approved exoskeleton, including the ReWalk P6.0. To the best of our knowledge, no prior study has reported a direct comparison of joint kinematics, ground reaction forces, EMG, and joint moments from a single session of controlled experiments with and without an FDA-approved exoskeleton. Our study advances previous work done with the ReWalk device in the following ways. First, three prior studies have quantified joint kinematics during exoskeletal-assisted walking in the ReWalk device [26, 29, 30]; our study builds on this prior work to provide a direct comparison of joint kinematics from exoskeletal-assisted and unassisted walking, sit-to-stand, and stand-to-sit activities in the ReWalk device (Fig 2, S1 and S5 Figs). Second, Hayes et al. reported ground reaction forces [29, 30] and Fineberg et al. reported foot reaction forces (obtained from in-shoe pressure insoles, F-Scan, TekScan, Boston, MA) during exoskeletal-assisted walking in the ReWalk device [14]; our study builds on this prior work to provide a direct comparison of ground reaction forces from exoskeletal-assisted and unassisted walking, sit-to-stand, and stand-to-sit activities in the ReWalk device (Fig 3, S2 and S6 Figs). Third, no prior study has reported EMG or joint moments from any activity in the ReWalk device; our EMG (Fig 4, S3 and S7 Figs) and joint moments (Fig 5, S4 and S8 Figs) results add to the body of knowledge and provide a minimum empirical dataset needed to quantify the dynamics of the human-robot system during exoskeletal-assisted and unassisted walking, sit-to-stand, and stand-to-sit activities in the ReWalk device.

We have also compared our findings with data that is available in the literature from other FDA-approved exoskeletons. Our joint kinematics results from exoskeletal-assisted walking in the ReWalk P6.0 are comparable to those reported by Chang et al. using the Ekso device [28]. Our mean peak hip extension, knee flexion, and ankle plantarflexion angles were 0.2˚-3.1˚,

50.1°-52.6°, and 6.7°-6.9°, respectively (Fig 2); in comparison, mean peak hip extension, knee flexion, and ankle plantarflexion angles reported in Chang et al. are ~3.0°-4.0°, ~48.0°-50.0°, and ~6.0°-9.0°, respectively [28]. Ekelem et al. reported median peak hip and knee flexion angles of ~2.0°-16.0° and ~23.0°-25.0°, respectively, during exoskeletal-assisted walking in the Indego device [27]. Because they do not report peak hip extension angles during stance, a direct comparison with our hip extension angle data is not possible. Their peak knee flexion values (~23.0°-25.0°) are lower than our values (50.1°-52.6°). This is likely because the two participants in their study had high levels of spasticity; with functional electrical stimulation, Ekelem et al. were able to increase the participants' peak knee flexion angles to ~35.0°-47.0° [27]. Next, Bellitto et al. compared EMG data from three incomplete spinal cord injured participants during exoskeletal-assisted (with the Ekso device) and unassisted walking [31]. The timings of our EMG activity are in agreement, but our normalized magnitudes are lower than those reported in Bellitto et al. This is likely because of two reasons: 1) our results are from an able-bodied participant, compared to incomplete spinal cord injured participants in Bellitto et al.; and 2) we normalized our EMG to maximum voluntary contraction data, while Bellitto et al. used a dynamic normalization technique [66], where EMG data from each muscle were normalized to the maximum value over all trials with and without the exoskeleton. There is no prior study with any FDA-approved exoskeletons to compare our results from sit-to-stand and stand-to-sit maneuvers.

This study provides a much-needed empirical dataset to develop virtual simulators reproducing EAL in FDA-approved devices. To the best of our knowledge, there is not a single virtual simulator of EAL in any FDA-approved exoskeleton. Prior studies have developed virtual simulators of EAL in custom-built exoskeletons [46–49, 51, 67], but these devices are not FDA-approved and are likely many years away from widespread application for the rehabilitation of patients with neurological disorders. Prior studies have also developed virtual simulators of EAL based on idealized or imaginary exoskeletons [39, 42–44, 50, 52]. For example, Bianco et al. developed a virtual simulator to examine how multi-joint assistance affects the metabolic cost of exoskeletal-assisted and unassisted walking [39]. This simulator, however, was not based on a physical exoskeleton and empirical data from unassisted walking was used as model inputs, with the assumption that joint kinematics, ground reaction forces, and EMG do not change between exoskeletal-assisted and unassisted walking. They stated in their limitations that incorporating the kinematic adaptations due to an exoskeleton would alter their metabolic cost predictions. It is likely that Bianco et al. and other studies that developed virtual simulators based on idealized or imaginary exoskeletal-assistance did so because they did not have empirical data from EAL.

A potential limitation of this study is that the ReWalk P6.0 was designed for patients with spinal cord injury, but the results from this study were obtained from a single able-bodied participant. We are currently actively recruiting able-bodied and spinal cord injured participants to validate the findings of our study. The goal of this manuscript is to report our novel framework combining controlled experiments with computational modeling to build subject-specific virtual simulators of EAL reproducing walking, sit-to-stand, and stand-to-sit maneuvers. Developing this framework represents substantial effort. We have described our methods in sufficient details for the research community to reproduce our work. In addition, we will make all empirical data from this study freely available through an online repository. There is benefit to publishing our dataset at the present time to enable the modeling community can start building on our work to address a broad range of research questions. A second limitation of this study is that our EAL experiments did not measure the external crutch reaction forces or the interaction forces due to the knee brackets and straps that were used to secure the participant in the exoskeleton. These forces were also not included in the virtual simulator and, as

such, their effects on our findings remain undefined. A third potential limitation is that we incorporated the mass and inertial properties of the exoskeleton with the musculoskeletal segments, similar to that reported in previous studies, and thus simplified our virtual simulator for ID analyses [62, 63]. A more thorough approach would have been to segregate the human and robot components, include human-robot interaction forces as input and to solve the human and robot kinematic chains separately.

This study provides a framework for determining the dynamics of the human-robot system during EAL in an FDA-approved exoskeleton using controlled experiments and a subject-specific virtual simulator. Our framework is applicable to exoskeletons that are not FDA-approved. We created a first-of-a-kind empirical dataset comprising joint kinematics, ground reaction forces, EMG, and joint moments from exoskeletal-assisted and unassisted locomotion. The virtual simulator will provide a low-risk and cost-effective platform for parametric studies of human-robot interaction during EAL, a necessary step for researchers and clinicians to characterize the effects of human factors on EAL and establish standards for safe and sustained use of robotic exoskeletons. Of practical significance, this virtual simulator model will provide a platform for exoskeleton companies to conduct rapid design-phase evaluations to accelerate device refinements. We invite investigators to build on our work to develop accurate virtual simulators reproducing EAL to address a broad range of questions, permitting the design and manufacture of exoskeletal devices that more closely replicate a normal gait pattern.

## Supporting information

**S1 Fig. Average (±1 SD) joint angles from four exoskeletal-assisted (A-F; human: blue for left leg, red for right leg; robot: black) and five unassisted (G-L; human: blue for left leg, red for right leg) sit-to-stand trials.** The offset in angles between the human (blue or red) and the robot (black) was due to different definitions of coordinate axes for each rigid body. (A-F) The first dashed vertical lines represent the transition from positioning phase (user gets in to position by flexing the torso forward and loading the crutches) to hold phase (leaned forward for 3 seconds). The second dashed vertical lines represent the beginning of the rise phase, with the hip and knee motors activating to complete the sit-to-stand maneuver. (TIF)

**S2 Fig. Average (±1 SD) ground reaction forces from four exoskeletal-assited (A-C) and five unassisted (D-F) sit-to-stand trials.** The ground reaction forces from exoskeletal-assisted locomotion were normalized to the combined weight of the participant and the exoskeleton; forces from unassisted locomotion were normalized to the participant's body weight (BW). (A-C) The first dashed vertical lines represent the transition from positioning phase (user gets in to position by flexing the torso forward and loading the crutches) to hold phase (leaned forward for 3 seconds). The second dashed vertical lines represent the beginning of the rise phase, with the hip and knee motors activating to complete the sit-to-stand maneuver. A-P = Anterior-Posterior, M-L = Medial-Lateral. (TIF)

**S3 Fig. Average (+1 SD) normalized electromyography (EMG) data from four exoskeletal-assisted (A-H) and five unassisted (I-P) sit-to-stand trials.** The EMG data were normalized using muscle-specific maximum voluntary contraction values. (A-H) The first dashed vertical lines represent the transition from positioning phase (user gets in to position by flexing the torso forward and loading the crutches) to hold phase (leaned forward for 3 seconds). The second dashed vertical lines represent the beginning of the rise phase, with the hip and knee

motors activating to complete the sit-to-stand maneuver. RF = rectus femoris, VL = vastus lateralis, VM = vastus medialis, ST = semitendinosus, BF = biceps femoris, GM = gastrocnemius medialis, SOL = soleus, TIB = tibialis anterior.
(TIF)

**S4 Fig. Average (±1 SD) joint moments from four exoskeletal-assited (A-F) and five unassisted (G-L) sit-to-stand trials.** The joint moments from exoskeletal-assisted locomotion were normalized to the combined mass of the participant and the exoskeleton; joint moments from unassisted locomotion were normalized to the participant's mass. (A-F) The first dashed vertical lines represent the transition from positioning phase (user gets in to position by flexing the torso forward and loading the crutches) to hold phase (leaned forward for 3 seconds). The second dashed vertical lines represent the beginning of the rise phase, with the hip and knee motors activating to complete the sit-to-stand maneuver.
(TIF)

**S5 Fig. Average (±1 SD) joint angles from five exoskeletal-assited (A-F; human: blue for left leg, red for right leg; robot: black) and five unassisted (G-L; human: blue for left leg, red for right leg) stand-to-sit trials.** The offset in angles between the human (blue or red) and the robot (black) was due to different definitions of coordinate axes for each rigid body. (A-F) The first dashed vertical lines represent the transition from positioning phase (user gets in to position by leaning backwards and loading the crutches) to hold phase (leaned backward for 3 seconds). The second dashed vertical lines represent the beginning of the descend phase, with the hip and knee motors activating to complete the stand-to-sit maneuver.
(TIF)

**S6 Fig. Average (±1 SD) ground reaction forces from five exoskeletal-assited (A-C) and five unassisted (D-F) stand-to-sit trials.** The ground reaction forces from exoskeletal-assisted locomotion were normalized to the combined weight of the participant and the exoskeleton; forces from unassisted locomotion were normalized to the participant's body weight (BW). (A-C) The first dashed vertical lines represent the transition from positioning phase (user gets in to position by leaning backwards and loading the crutches) to hold phase (leaned backward for 3 seconds). The second dashed vertical lines represent the beginning of the descend phase, with the hip and knee motors activating to complete the stand-to-sit maneuver.
A-P = Anterior-Posterior, M-L = Medial-Lateral.
(TIF)

**S7 Fig. Average (+1 SD) normalized electromyography (EMG) data from five exoskeletal-assisted (A-H) and five unassisted (I-P) stand-to-sit trials.** The EMG data were normalized using muscle-specific maximum voluntary contraction values. (A-H) The first dashed vertical lines represent the transition from positioning phase (user gets in to position by leaning backwards and loading the crutches) to hold phase (leaned backward for 3 seconds). The second dashed vertical lines represent the beginning of the descend phase, with the hip and knee motors activating to complete the stand-to-sit maneuver. RF = rectus femoris, VL = vastus lateralis, VM = vastus medialis, ST = semitendinosus, BF = biceps femoris, GM = gastrocnemius medialis, SOL = soleus, TIB = tibialis anterior.
(TIF)

**S8 Fig. Average (±1 SD) joint moments from five exoskeletal-assited (A-F) and five unassisted (G-L) stand-to-sit trials.** The joint moments from exoskeletal-assisted locomotion were normalized to the combined mass of the participant and the exoskeleton; joint moments from unassisted locomotion were normalized to the participant's mass. (A-F) The first dashed

vertical lines represent the transition from positioning phase (user gets in to position by leaning backwards and loading the crutches) to hold phase (leaned backward for 3 seconds). The second dashed vertical lines represent the beginning of the descend phase, with the hip and knee motors activating to complete the stand-to-sit maneuver.
(TIF)

## Acknowledgments

We thank Assaf Tzioni from ReWalk Robotics for his support with access to the exoskeleton encoder data. We thank Lina Alsauskaite from ReWalk Robotics for training the research team on safe use of the exoskeleton. ReWalk Robotics did not play any role in study design, participant recruitment, methodology, and data analysis.

## Author Contributions

**Conceptualization:** Vishnu D. Chandran, William A. Bauman, Saikat Pal.

**Data curation:** Vishnu D. Chandran.

**Formal analysis:** Vishnu D. Chandran, William A. Bauman, Saikat Pal.

**Funding acquisition:** William A. Bauman, Saikat Pal.

**Investigation:** Vishnu D. Chandran, Sanghyun Nam, Saikat Pal.

**Methodology:** Vishnu D. Chandran, Sanghyun Nam, Saikat Pal.

**Project administration:** Saikat Pal.

**Resources:** David Hexner, William A. Bauman, Saikat Pal.

**Software:** Vishnu D. Chandran, Sanghyun Nam.

**Supervision:** Saikat Pal.

**Validation:** Saikat Pal.

**Visualization:** Vishnu D. Chandran.

**Writing – original draft:** Saikat Pal.

**Writing – review & editing:** Vishnu D. Chandran, Sanghyun Nam, David Hexner, William A. Bauman, Saikat Pal.

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
