## [Decision Letter · Decision Letter 0]

29 Nov 2022

PONE-D-22-15991Comparison of the dynamics of exoskeletal-assisted and unassisted locomotion in an FDA-approved lower extremity device: Controlled experiments and development of a subject-specific virtual simulatorPLOS ONE

Dear Dr. Pal,

Thank you for submitting your manuscript to PLOS ONE. After careful consideration, we feel that it has merit but does not fully meet PLOS ONE’s publication criteria as it currently stands. Therefore, we invite you to submit a revised version of the manuscript that addresses the points raised during the review process.

We look forward to receiving your revised manuscript.

Kind regards,

Imre Cikajlo, Ph.D.

Academic Editor

PLOS ONE

Journal Requirements:

2. Thank you for providing the following information in your Ethics Statement: "The Institutional Review Board at New Jersey Institute of Technology approved this study. The approval number is 2008001531R001." We ask that you also amend your Methods section to include this statement.

“WAB and SP received support from the Department of Veterans Affairs (https://www.va.gov/) grant VA RR&D # 1 I01 RX003561-01A2. The funders had no role in study design, data collection and analysis, decision to publish, or preparation of the manuscript.”

Reviewers' comments:

Reviewer's Responses to Questions

**Comments to the Author**

1. Is the manuscript technically sound, and do the data support the conclusions?

Reviewer #1: Partly

Reviewer #2: No

2. Has the statistical analysis been performed appropriately and rigorously? 

Reviewer #1: N/A

Reviewer #2: N/A

3. Have the authors made all data underlying the findings in their manuscript fully available?

Reviewer #1: Yes

Reviewer #2: Yes

4. Is the manuscript presented in an intelligible fashion and written in standard English?

Reviewer #1: Yes

Reviewer #2: Yes

5. Review Comments to the Author

Reviewer #1: This is a well-written paper that describes an experiment and simulator with one subject performing walking and sit-to-stand motions with and without an exoskeleton. I believe the manuscript is eventually worthy of publication, though I have a number of major and minor comments.

MAJOR REVISION

1. The authors repeatedly emphasize that existing studies have not been performed on an FDA-approved exoskeleton, and that the novelty of the work is usage with an FDA-approved exoskeleton. However, it is unclear why the FDA approval represents a major difference. It seems like existing studies have achieved similar results with less popular exoskeletons, and most of the approaches used in existing studies can be approved to other exoskeletons with relatively little modification. The authors should better explain why the study is more than simply applying existing methods to a more popular exoskeleton.

2. Related to the above: the Discussion acknowledges that several studies have already been done with the ReWalk exoskeleton (which the authors are using). The authors should better characterize how their work advances previous ReWalk work rather than simply saying their results are in line with previous ReWalk results.

3. Why did the authors recruit only a single subject for a single session? It seems like there would be significant intersubject variability that would both make it difficult to compare exoskeleton vs. no-exoskeleton conditions as well as make it difficult to evaluate performance of a simulation model. Additionally, since this was an able-bodied subject who only participated for a few sessions, it seems like the authors could have fairly easily obtained a larger sample size.

4. The simulation model does not appear to be extensively validated - the authors show that it can fairly accurately reproduce actual trajectories, but this is done by training and testing on the same subject in the same session. Is this sufficient for a real simulator? I would have expected it to be validated by examining predictive ability in different situations. And when the authors say that the simulator reproduced actual trajectories "within acceptable tolerances", what is an acceptable tolerance?

MINOR REVISION

1. Is it really necessary to list inclusion/exclusion criteria when only a single participant was recruited? Authors list things like pregnancy and lactation being exclusion criteria, which seems irrelevant for a male participant. They also list minimum and maximum weight, which again seems irrelevant since a single participant was recruited and his weight is known.

2. Could the authors briefly describe how maximum voluntary contraction protocols were performed?

3. Why were there 4 trials of exoskeleton-assisted sit-to-stand and 5 trials of exoskeleton-unassisted sit-to-stand? Why not 5 for both?

Reviewer #2: The manuscript presents a dataset collected on a healthy subject performing exoskeleton-assisted vs unassisted locomotion using an FDA-approved exoskeleton (ReWalk P6.0), and a subject-specific virtual simulator implemented using OpenSim environment, to compute joint kinematics (from motion capture and simulation) and moments (from simulation) for hip, knee and ankle in assisted vs unassisted conditions. in addition, EMG data from 8 lower-limb muscles were recorded, as well as ground reaction forces using force plates.

The main novelty claimed by the authors is the availability of such a dataset and simulation that includes an FDA-approved exoskeleton comparing assisted and unassisted locomotion. Nevertheless, whereas the need for having such a dataset and simulation is well explained in the manuscript, it is not clear which is the added value of including an FDA-approved exoskeleton, with respect to other examples (cited by the authors in the manuscript) using different exoskeletons. In my opinion, this alone cannot be the main claim of the manuscript given that: no novel methods are presented; different exoskeletons might result in different human-robot interaction when assistance is delivered; other similar data in literature could be used for the simulator if the benefit would be the possibility to "conduct rapid design-phase evaluations to accelerate device refinements" (Discussion).

The authors have stated clearly the limitations of the study, including having no participants with altered gait patterns (e.g., spinal cord) and not separating human and robotic dynamics in the virtual simulator (one aspect that was described in the Introduction as particularly important and not very well explored in the state of the art). Exploring further more at least one of these limitations would be important to justify the significance of the manuscript. Indeed, in the current state, comparison with other studies involving impaired subjects (e.g., in the discussion) has a questionable significance.

Other comments:

- Kinematics' results are reported in the Figures without the offset, but absolute values of data without removing the offset are comparing in the manuscript. How should these differences be interpreted? Comparing and presenting directly data without offset should be more straightforward.

- The standard deviation values of the RMS errors between experiments and simulation should be reported together with the mean values.

6. PLOS authors have the option to publish the peer review history of their article (what does this mean?). If published, this will include your full peer review and any attached files.

Reviewer #1: No

Reviewer #2: No

---

## [Author Response · Author response to Decision Letter 0]

6 Dec 2022

Please see our Response to Reviewers document for proper formatting of the text below. 

We thank the editor and the reviewers for their thorough review of our work. Responding to the reviewers’ comments has improved this manuscript. The specific responses to the comments are described below, and the changes are highlighted in the revised manuscript. Please note that the sentences in italics are reviewer comments.

Journal Requirements:

1. Please ensure that your manuscript meets PLOS ONE's style requirements, including those for file naming. The PLOSONE style templates can be found at

Authors’ Response: We have modified the manuscript to comply with style requirements for PLOS ONE. 

2. Thank you for providing the following information in your Ethics Statement: "The Institutional Review Board at New Jersey Institute of Technology approved this study. The approval number is 2008001531R001." We ask that you also amend your Methods section to include this statement.

Authors’ Response: We added the requested sentence in the Methods section. 

“WAB and SP received support from the Department of Veterans Affairs (https://www.va.gov/) grant VA RR&D # 1 I01RX003561-01A2. The funders had no role in study design, data collection and analysis, decision to publish, or preparation of the manuscript.”

Authors’ Response: We included an amended Funding Statement in our cover letter, as requested. 

Reviewers' Comments:

5. Review Comments to the Author

Reviewer #1

This is a well-written paper that describes an experiment and simulator with one subject performing walking and sit-to-stand motions with and without an exoskeleton. I believe the manuscript is eventually worthy of publication, though I have a number of major and minor comments.

Authors’ Response: We thank the reviewer for their positive feedback. 

MAJOR REVISION

1. The authors repeatedly emphasize that existing studies have not been performed on an FDA-approved exoskeleton, and that the novelty of the work is usage with an FDA-approved exoskeleton. However, it is unclear why the FDA approval represents a major difference. It seems like existing studies have achieved similar results with less popular exoskeletons, and most of the approaches used in existing studies can be approved to other exoskeletons with relatively little modification. The authors should better explain why the study is more than simply applying existing methods to a more popular exoskeleton.

Authors’ Response: We thank the reviewer for the opportunity to clarify why we restricted our work to an FDA-approved exoskeleton device. It is important to study the dynamics of the human-robot system during exoskeletal-assisted locomotion (EAL) in FDA-approved devices because individuals with neurological disorders only have access to FDA-approved devices. Currently, there is a disconnect between the exoskeletons used by patients and exoskeletons used to study human-robot interaction. Our study is a first step to address this disconnect. We have added several sentences in Introduction to highlight the importance of studying the dynamics of exoskeleton-assisted locomotion in FDA-approved devices. 

 Next, prior studies with FDA-approved exoskeletons have quantified joint kinematics, EMG, foot reaction forces, and ground reaction forces. To the best of our knowledge, no prior study has reported joint moments of the human-robot system during EAL in an FDA-approved device. We have clarified this gap in knowledge in Introduction section. 

2. Related to the above: the Discussion acknowledges that several studies have already been done with the ReWalk exoskeleton (which the authors are using). The authors should better characterize how their work advances previous ReWalk work rather than simply saying their results are in line with previous ReWalk results.

Authors’ Response: We have added a paragraph in the Discussion section to highlight how our work advances previous work performed with the ReWalk exoskeleton, as requested.

3. Why did the authors recruit only a single subject for a single session? It seems like there would be significant intersubject variability that would both make it difficult to compare exoskeleton vs. no-exoskeleton conditions as well as make it difficult to evaluate performance of a simulation model. Additionally, since this was an able-bodied subject who only participated for a few sessions, it seems like the authors could have fairly easily obtained a larger sample size.

Authors’ Response: We agree with the Reviewer that there will be some inter-subject variability and additional participants will be required to validate the findings of this study. We respectfully disagree that adding additional participants to our study would be easy. To this point, conducting the controlled experiments and developing the subject-specific virtual simulators of walking, sit-to-stand, and stand-to-sit maneuvers with and without the exoskeleton are extremely laborious and time-consuming tasks. We currently are actively recruiting 7 able-bodied and 10 spinal cord injured participants to extend this work and anticipate that it will take at least a couple of years to recruit, study, and complete the analyses on these participants. 

To clarify, the goal of this manuscript is to report our novel framework combining controlled experiments with computational modeling to build subject-specific virtual simulators of EAL that reproduces walking, sit-to-stand, and stand-to-sit maneuvers. Developing this framework represents substantial effort. We have described our methods in sufficient detail for the research community to reproduce our work. In addition, we will make all empirical data from this study freely available to the research community. There is benefit to publishing our dataset now so the modeling community can start building on our work to address a broad range of research questions. We have added several sentences in the section that addresses study limitations to clarify this point.

4. The simulation model does not appear to be extensively validated - the authors show that it can fairly accurately reproduce actual trajectories, but this is done by training and testing on the same subject in the same session. Is this sufficient for a real simulator? I would have expected it to be validated by examining predictive ability in different situations. And when the authors say that the simulator reproduced actual trajectories "within acceptable tolerances", what is an acceptable tolerance?

Authors’ Response: We agree that we have not yet tested the predictive capability of our virtual simulators. As we have stated previously in our response, the goal of this study was to develop the framework to build subject-specific virtual simulators of EAL. Testing the predictive capability of these virtual simulators will be a component of our future work. We invite other investigators to use our dataset to test the predictive capability of our virtual simulators, or to build their own virtual simulators from our empirical data. We have clarified in the Abstract that this work provides a foundation for parametric studies to characterize the effects of human and robot design variables, and predictive modeling to optimize human-robot interaction during EAL. We have also added a few sentences at the end of the Discussion section encouraging other investigators to build on our work. 

 Next, acceptable tolerance is less than 2 cm average RMS error between experiment and simulator markers, per OpenSim’s best practices, which has been added with a sentence and two references in the Results section.

MINOR REVISION

1. Is it really necessary to list inclusion/exclusion criteria when only a single participant was recruited? Authors list things like pregnancy and lactation being exclusion criteria, which seems irrelevant for a male participant. They also list minimum and maximum weight, which again seems irrelevant since a single participant was recruited and his weight is known.

Authors’ Response: We thank the reviewer for this comment and in response have removed the inclusion/exclusion criteria from the manuscript. 

2. Could the authors briefly describe how maximum voluntary contraction protocols were performed?

Authors’ Response: We have added several sentences in the Methods section to describe our maximum voluntary contraction trials. 

3. Why were there 4 trials of exoskeleton-assisted sit-to-stand and 5 trials of exoskeleton-unassisted sit-to-stand? Why not 5 for both?

Authors’ Response: We collected 10 trials each of walking, sit-to-stand, and stand-to-sit maneuvers, but only successful trials were included for further analyses. We have clarified in the Methods section the criteria for a successful trial for each maneuver. Meeting these criteria resulted in different numbers of trials for the different maneuvers. Based on these criteria, we obtained 6 successful trials each of exoskeletal-assisted and unassisted walking, 4 and 5 successful trials of exoskeletal-assisted and unassisted sit-to-stand maneuvers, respectively, and 5 successful trials each of exoskeletal-assisted and unassisted stand-to-sit maneuvers. 

Reviewer #2

The manuscript presents a dataset collected on a healthy subject performing exoskeleton-assisted vs unassisted locomotion using an FDA-approved exoskeleton (ReWalk P6.0), and a subject-specific virtual simulator implemented using OpenSim environment, to compute joint kinematics (from motion capture and simulation) and moments (from simulation) for hip, knee and ankle in assisted vs unassisted conditions. In addition, EMG data from 8 lower-limb muscles were recorded, as well as ground reaction forces using force plates.

The main novelty claimed by the authors is the availability of such a dataset and simulation that includes an FDA-approved exoskeleton comparing assisted and unassisted locomotion. Nevertheless, whereas the need for having such a dataset and simulation is well explained in the manuscript, it is not clear which is the added value of including an FDA-approved exoskeleton, with respect to other examples (cited by the authors in the manuscript) using different exoskeletons. In my opinion, this alone cannot be the main claim of the manuscript given that: no novel methods are presented; different exoskeletons might result in different human-robot interaction when assistance is delivered; other similar data in literature could be used for the simulator if the benefit would be the possibility to "conduct rapid design-phase evaluations to accelerate device refinements" (Discussion).

Authors’ Response: We thank the reviewer for this comment and have addressed the importance of studying the dynamics of exoskeleton-assisted locomotion in FDA-approved devices in Reviewer 1’s first major revision comment. To briefly reiterate our prior response, it is important to study the dynamics of the human-robot system during EAL in FDA-approved devices because individuals with neurological disorders only have access to FDA-approved devices. Currently, there is a disconnect between the exoskeletons used by patients and exoskeletons used to study human-robot interaction. Our study is a first step to address this disconnect. We have added several sentences in the Introduction section to highlight the importance of studying the dynamics of exoskeleton-assisted locomotion in FDA-approved devices. 

Next, our framework that combines controlled experiments with computational modeling to build subject-specific virtual simulators of EAL is novel. No prior study has simulated EAL in any FDA-approved exoskeleton. The minimum empirical dataset we will share as part of this publication is novel. There is no such dataset currently available to the research community, which explains why the computational modeling community has developed virtual simulators based on idealized or imaginary exoskeletons (Bianco et al., 2022; Dembia et al., 2017; Khamar et al., 2019; Nguyen et al., 2019; Uchida et al., 2016; Zhu et al., 2015). Furthermore, we have added two paragraphs in the Discussion section to highlight how our work advances previous work performed with the ReWalk and other FDA-approved exoskeletons.

The authors have stated clearly the limitations of the study, including having no participants with altered gait patterns (e.g., spinal cord) and not separating human and robotic dynamics in the virtual simulator (one aspect that was described in the Introduction as particularly important and not very well explored in the state of the art). Exploring further more at least one of these limitations would be important to justify the significance of the manuscript. Indeed, in the current state, comparison with other studies involving impaired subjects (e.g., in the discussion) has a questionable significance.

Authors’ Response: We thank the reviewer for this comment and have modified our text to provide a more detailed discussion of the limitation of including a single able-bodied participant in this study. Next, we have modified the text in the Discussion section to clarify how our work advances previous work done with the ReWalk and other FDA-approved exoskeletons.

Other comments:

- Kinematics' results are reported in the Figures without the offset, but absolute values of data without removing the offset are comparing in the manuscript. How should these differences be interpreted? Comparing and presenting directly data without offset should be more straightforward.

Authors’ Response: This comment is somewhat unclear to the authors. However, in our attempt to clarify, our segment coordinate frames were obtained with reference to a single global coordinate frame. The initial values of the joint kinematics from the different segment coordinate frames were based on this single global coordinate frame. This provided a direct comparison of joint kinematics with and without the exoskeleton. For example, using this approach we clearly see that the participant’s hips were more flexed during exoskeletal-assisted walking compared to unassisted walking (Figs 2A and 2B). If we were to remove the offset angles between the different segment coordinate axes, this would mask many of the kinematic differences that are of interest when analyzing locomotion with and without the exoskeleton. In our results, we have only subtracted the offset angles to calculate the average absolute differences in kinematics between the human and the robot during exoskeletal-assisted walking (Figs 2A-2D). To clarify the rationale for the approach taken, text in the Results section has been appropriately edited. 

- The standard deviation values of the RMS errors between experiments and simulation should be reported together with the mean values.

Authors’ Response: We have included the standard deviation values of the RMS errors between experiment and simulator markers, as requested.

---

## [Decision Letter · Decision Letter 1]

8 Jan 2023

PONE-D-22-15991R1Comparison of the dynamics of exoskeletal-assisted and unassisted locomotion in an FDA-approved lower extremity device: Controlled experiments and development of a subject-specific virtual simulatorPLOS ONE

Dear Dr. Pal,

Thank you for submitting your manuscript to PLOS ONE. After careful consideration, we feel that it has merit but does not fully meet PLOS ONE’s publication criteria as it currently stands. Therefore, we invite you to submit a revised version of the manuscript that addresses the points raised during the review process.

We look forward to receiving your revised manuscript.

Kind regards,

Imre Cikajlo, Ph.D.

Academic Editor

PLOS ONE

Journal Requirements:

Additional Editor Comments:

Please pay attention to the issues raised by the reviewer 1.

Reviewers' comments:

Reviewer's Responses to Questions

**Comments to the Author**

1. If the authors have adequately addressed your comments raised in a previous round of review and you feel that this manuscript is now acceptable for publication, you may indicate that here to bypass the “Comments to the Author” section, enter your conflict of interest statement in the “Confidential to Editor” section, and submit your "Accept" recommendation.

Reviewer #1: (No Response)

Reviewer #2: All comments have been addressed

2. Is the manuscript technically sound, and do the data support the conclusions?

Reviewer #1: Partly

Reviewer #2: Yes

3. Has the statistical analysis been performed appropriately and rigorously? 

Reviewer #1: Yes

Reviewer #2: N/A

4. Have the authors made all data underlying the findings in their manuscript fully available?

Reviewer #1: No

Reviewer #2: No

5. Is the manuscript presented in an intelligible fashion and written in standard English?

Reviewer #1: Yes

Reviewer #2: Yes

6. Review Comments to the Author

Reviewer #1: The authors have partially addressed my comments from the first round of review, and I believe the paper is now somewhat closer to publication. However, I have several remaining comments and thus recommend another round of major revisions.

MAJOR ISSUES

1. The authors repeatedly emphasize that the novel dataset collected during the study would be of value to researchers. However, as far as I can tell, they have not made the dataset available, and present conflicting statements in the PLOS ONE submission system. The Data Availability box states "Yes - all data are fully available without restriction", but the "Describe where the data may be found in full sentences" box then simply states "All empirical data from this study will be freely available." The Discussion also states that all data "will be" fully available. If the novel dataset is part of the value of the manuscript, the authors should make it available during the review process. If that is not possible for some exceptional reason, the authors should at least provide some indication of how the data will be made available so that the approach can be evaluated.

2. The authors state that, to their knowledge, no previous study has reported joint moments. I would appreciate a brief explanation of how difficult it is to obtain these joint moments and what the usefulness of joint moments is relative to other data that was already previously done. This would help evaluate the degree of methodological novelty.

3. I agree that patients have access only to FDA-approved devices, but at the same time, the authors claim that the goal of their work is to "report our novel framework combining controlled experiments with computational modeling to build subject-specific virtual simulators". A novel framework should be applicable to many exoskeletons, not just the specific FDA-approved one, and it is thus still not convincing why the authors emphasize FDA approval so much. Is the novelty that this is the first empirical dataset of this scope with an FDA-approved device?

4. The authors' justification for why only 1 subject is possible is that it would take years to conduct a study with ~20 participants. If development of virtual simulators for specific individuals is so time-consuming (seemingly on the scale of months / participant), the authors should better justify the expected practical value of such simulators. For real patients, economic and logistical issues would likely not allow months of time to be devoted to tuning a simulator to each person.

MINOR ISSUES

1. The authors' justification for why it is important to study the dynamics of the human-robot system during EAL in FDA-approved devices appears to have been awkwardly dropped into the Introduction without much effort to make it "flow" well with the rest of the text. Would recommend better integrating with previously existing text.

2. Lines 171-172: "A generic musculoskeletal [55] was scaled to the participant’s anthropometry." A noun seems to be missing here.

3. The authors state that they conducted 10 trials for each activity and then analyzed only the successful trials, leading to different numbers of trials in each activity. Why did the authors alternatively not repeat trials until a certain number of successful trials was obtained? Is it not possible to check for missing data within the session?

4. The Introduction says that the new dataset is novel because it includes joint moments, which had not been previously available. The Discussion then says that the new dataset is novel because it includes joint moments and EMG. If the EMG is also novel, I recommend mentioning that in the Introduction as well.

Reviewer #2: The authors have addressed the Reviewers' comments and the contribution of the manuscript is more clear.

I would suggest to carefully check the new text that was included to avoid repetitions (i.e., the sentence "we will make all empirical data from this study freely available to the research community" is repeated twice in the Discussion). Regarding this point, it would be more sound to write in the manuscript that the dataset is already available (instead of "will be available")

7. PLOS authors have the option to publish the peer review history of their article (what does this mean?). If published, this will include your full peer review and any attached files.

Reviewer #1: No

Reviewer #2: No

---

## [Author Response · Author response to Decision Letter 1]

10 Jan 2023

Please see the Response to Reviewers.pdf document. Thank you.

---

## [Decision Letter · Decision Letter 2]

25 Jan 2023

Comparison of the dynamics of exoskeletal-assisted and unassisted locomotion in an FDA-approved lower extremity device: Controlled experiments and development of a subject-specific virtual simulator

PONE-D-22-15991R2

Dear Dr. Pal,

We’re pleased to inform you that your manuscript has been judged scientifically suitable for publication and will be formally accepted for publication once it meets all outstanding technical requirements.

Kind regards,

Imre Cikajlo, Ph.D.

Academic Editor

PLOS ONE

Additional Editor Comments (optional):

Reviewers' comments:

Reviewer's Responses to Questions

**Comments to the Author**

1. If the authors have adequately addressed your comments raised in a previous round of review and you feel that this manuscript is now acceptable for publication, you may indicate that here to bypass the “Comments to the Author” section, enter your conflict of interest statement in the “Confidential to Editor” section, and submit your "Accept" recommendation.

Reviewer #1: (No Response)

2. Is the manuscript technically sound, and do the data support the conclusions?

Reviewer #1: Yes

3. Has the statistical analysis been performed appropriately and rigorously? 

Reviewer #1: Yes

4. Have the authors made all data underlying the findings in their manuscript fully available?

Reviewer #1: No

5. Is the manuscript presented in an intelligible fashion and written in standard English?

Reviewer #1: Yes

6. Review Comments to the Author

Reviewer #1: The authors have adequately addressed my comments, and I am fine with the paper being accepted as-is. I would recommend discretionary changes:

Consider making changes to the text in response to my last major issue #4 and minor issue #3. While the authors' response (in the Response to Reviewer document) is adequate, the authors do not appear to have changed the text in response, and other readers may have the same questions. For reference, the previously raised issues were:

- The authors' justification for why only 1 subject is possible is that it would take years to conduct

a study with ~20 participants. If development of virtual simulators for specific individuals is so

time-consuming (seemingly on the scale of months / participant), the authors should better justify

the expected practical value of such simulators. For real patients, economic and logistical issues

would likely not allow months of time to be devoted to tuning a simulator to each person.

- The authors state that they conducted 10 trials for each activity and then analyzed only the

successful trials, leading to different numbers of trials in each activity. Why did the authors

alternatively not repeat trials until a certain number of successful trials was obtained? Is it not

possible to check for missing data within the session?

7. PLOS authors have the option to publish the peer review history of their article (what does this mean?). If published, this will include your full peer review and any attached files.

Reviewer #1: No

---

## [Editor Report · Acceptance letter]

2 Feb 2023

PONE-D-22-15991R2 

Comparison of the dynamics of exoskeletal-assisted and unassisted locomotion in an FDA-approved lower extremity device: Controlled experiments and development of a subject-specific virtual simulator 

Dear Dr. Pal:

I'm pleased to inform you that your manuscript has been deemed suitable for publication in PLOS ONE. Congratulations! Your manuscript is now with our production department. 

Kind regards, 

on behalf of

Professor Imre Cikajlo 

Academic Editor

PLOS ONE